

# A survey of field programmable gate array (FPGA)-based graph convolutional neural network accelerators: challenges and opportunities

Shun Li[1,*], Yuxuan Tao[2,*], Enhao Tang[1], Ting Xie[1] and Ruiqi Chen[3,4]

[1] College of Physics and Information Engineering, Fuzhou University, Fuzhou, Fujian, China
[2] Department of Informatics Faculty of Natural, Mathematical & Engineering Sciences, King's College London, Strand, London, United Kingdom
[3] VeriMake Innovation Lab, Nanjing Renmian Integrated Circuit Co., Ltd., Nanjing, Jiangsu, China
[4] Zhangjiang Fudan International Innovation Center, Fudan University, Shanghai, Shanghai, China
* These authors contributed equally to this work.

Corresponding author
Ruiqi Chen,
rickychen@verimake.com

## ABSTRACT

Graph convolutional networks (GCNs) based on convolutional operations have been developed recently to extract high-level representations from graph data. They have shown advantages in many critical applications, such as recommendation system, natural language processing, and prediction of chemical reactivity. The problem for the GCN is that its target applications generally pose stringent constraints on latency and energy efficiency. Several studies have demonstrated that field programmable gate array (FPGA)-based GCNs accelerators, which balance high performance and low power consumption, can continue to achieve orders-of-magnitude improvements in the inference of GCNs models. However, there still are many challenges in customizing FPGA-based accelerators for GCNs. It is necessary to sort out the current solutions to these challenges for further research. For this purpose, we first summarize the four challenges in FPGA-based GCNs accelerators. Then we introduce the process of the typical GNN algorithm and several examples of representative GCNs. Next, we review the FPGA-based GCNs accelerators in recent years and introduce their design details according to different challenges. Moreover, we compare the key metrics of these accelerators, including resource utilization, performance, and power consumption. Finally, we anticipate the future challenges and directions for FPGA-based GCNs accelerators: algorithm and hardware co-design, efficient task scheduling, higher generality, and faster development.

# INTRODUCTION

Inspired by the powerful learning ability of neural networks and the great success of convolutional neural networks (CNNs) (*LeCun et al., 1998*) in the field of deep learning, graph neural networks (GCNs) based on convolutional operations such as GCN (*Bruna et al., 2013*), GraphSAGE (*Hamilton, Ying & Leskovec, 2017*), and GAT (*Veličković et al.,*

*2017*) were developed and used to extract high-level representations from graph data (*Wu et al., 2021*). GCN uses convolutional operations to learn node features, GraphSAGE uses neighborhood sampling to implement inductive learning on large-scale datasets, and GAT uses an attention mechanism to obtain the weight of neighbor nodes. These models have been successfully applied to many applications, such as social networks (*Wu et al., 2022*), knowledge graphs (*Arora, 2020*), and molecular attribute prediction (*Wieder et al., 2020*), and have gradually become a new addition to the data centers of many companies, such as Google, Facebook, and Alibaba. Despite the diversity of these models, in general, the computational process of GCNs can be roughly divided into two stages: aggregation and combination (*Abadal et al., 2021*). The irregular distribution of the number and location of neighbor nodes will cause the matrix to exhibit sparsity and irregularity, which seriously affects the inference speed of GCNs models. Accelerators based on software frameworks such as PyG (*Fey & Lenssen, 2019*) and DGL (*Wang et al., 2020*) simplify the execution of these two stages, but the improvement they bring is limited, so the efficient computation of GCNs has become a hot topic.

It is currently a popular and effective method to design accelerators for corresponding GCNs models using FPGA that take into account fine-grained computation, high parallelism, and programmability, such as AWB-GCN (*Geng et al., 2020*), LW-GCN (*Tao et al., 2021*), FPGAN (*Yan, Tong & Zhi, 2020*), BoostGCN (*Zhang, Kannan & Prasanna, 2021*), I-GCN (*Geng et al., 2021b*), *etc*. These customized FPGA models all completed the inference of GCNs efficiently through specific optimization.

However, implementing efficient inference of GCNs on FPGA is not a simple task. Due to the particularity of graph data, customizing FPGA accelerators for GCNs has the following challenges:

- **Efficient processing of sparse matrix**

  The inference process of neural networks is full of large matrix operations, which have different sparsity, and can lead to irregular memory accesses. The inefficiency of matrix operations will seriously affect the speed of model inference, and how to effectively resolve sparsity and data reuse is critical for efficient processing of sparse matrix.

- **Unbalanced workload**

  Since graph data has a different sparsity, as well as the fact that the memory location of the neighbors of each node and the number of neighbors of each node are irregular, it will result in an unbalanced workload among the nodes of the graph, thus reducing the computational efficiency.

- **Execution order differences**

  There are two steps in the GCNs model: aggregation and combination. The aggregation phase collects neighbor node information, and the combination phase completes the feature update. The combination phase relative to the aggregation phase can be considered a rule calculation.

- **Quantification and preservation of accuracy**

Compared with full-precision computing, fixed-point computing can significantly improve the speed of inference, but it will bring a certain loss of precision. At the same time, maintaining accuracy is very challenging when many optimizations are used in the model.

It is necessary to summarize the methods to deal with these challenges. We conduct a comprehensive survey of the current GCNs accelerators based on FPGA, which includes the design details of some accelerators under different challenges. We also analyze future development opportunities and provide guidance value for follow-up research work. Hopefully, people who are interested in the FPGA-based GCN accelerator design can benefit from this survey.

Our article has some significant contributions, which are summarized as follows:

1. To our knowledge, this is the first survey of current FPGA-based inference accelerators for GCNs. We list the current accelerators with excellent performance, introduce their characteristics and compare their performance, and introduce the details of some designs according to different challenges.
2. We review three famous GCNs models based on convolutional operations: GCN, GraphSAGE, and GAT. There are many GCNs based on convolution operations. In this article, we detail the inference process of three representative models.
3. We look forward to the future development direction and challenges of FPGA-based GCNs accelerators. The complexity of graph data will continuously challenge the acceleration of GCNs, and accelerators of software and hardware co-design can often maximize performance. Due to the unbalanced development between the algorithms and accelerators of GCNs, maintaining generality and accelerating the development speed are significant challenges for future FPGA-based GCNs accelerators.

The rest of this article is organized as follows. The "Survey methodology" will briefly introduce the development process and computing characteristics of GCNs, the advantages of FPGAs compared to CPUs and GPUs, and the sorting out of previous investigations on GNNs. "Background" introduces the traditional GNN model and several representative GCNs models. "GNN and GCNs models" lists the current state-of-the-art accelerators, presents their characteristics, details some designs according to four different challenges, and discusses their performance. "FPGA based hardware accelerators" summarizes this article and looks forward to the future development direction and challenges of FPGA-based GCNs accelerators.

## SURVEY METHODOLOGY

There are large volumes of non-Euclidean data produced in software applications, that are denoted to graphs with complex dependencies. These graphs pose challenges in efficient computing and modeling. The convolution-based GNNs are developed to extract hidden relations from the data and get superiority in graph representation learning. However, this

**Table 1 Search strategy.**

| Database | Initial search strategy | Updated search strategy |
|---|---|---|
| IEEE Xplore digital library | ("Full Text Only": FPGA) AND | ("All Metadata": FPGA) AND |
| | ("Full Text Only": hardware) OR | ("Full Text & Metadata": GCN) OR |
| | ("Full Text Only": software) AND | ("Full Text & Metadata": GNN) AND |
| | ("All Metadata": GCN) AND | ("Full Text & Metadata": accelerat) |
| | ("All Metadata": accelarator) | |
| ACM digital libraries | [All: GCN] AND | [All: GCN] OR |
| | [All: FPGA] AND | [All: GNN] AND |
| | | [All: FPGA] AND |
| | [All: Accelerator] | [All: accelerat] |
| | [Abstract: GCN] AND | [Abstract: GNN] OR |
| ArXiv | [Fulltext: FPGA] AND | [Abstract: GCN] AND |
| | [Fulltext: Accelerator] | [Abstract: FPGA] AND |
| | | [Fulltext: accelerat] |

results in a significant increase in computation time. Consequently, researchers began to pay attention to designing the GCN specifical accelerator. Unlike GPU and ASIC with fixed hardware architectures, FPGA is reconfigurable hardware, which means developers can connect the logical blocks within the FPGA through programmable connections to achieve their desired function (*Nurvitadhi et al., 2016*). In the design process of FPGA-based GCN accelerators, some challenges are presented or solved. However, the existing GCNs surveys don't focus on it. Before conducting it, the literature was needed to search and review. First, we chose the related electronic bibliographic databases such as IEEE Xplore and ACM Digital Libraries. Moreover, the arXiv is selected. Although the quality of studies is heterogeneous in arXiv, there is the newest research to be released. Next, we formulated a search strategy as illustrated in Table 1. After the first-round search, the keyword and search strategy were updated based on the keyword of results. Then, we make the second-round search. After it, we used Google Scholar to scan the references, cited in these articles with the snowballing approach (*Dengel et al., 2022*). It is worthwhile to mention that we focused on the FPGA-based designs published in the top FPGA conferences (FPGA, FCCM, FPL, FPT), EDA conferences (DAC, ASP-DAC, DATE, ICCAD), and architecture conferences (MICRO, HPCA, ISCA, ASPLOS) since 2019 (*Guo et al., 2019*). Because these articles are state-of-the-art in this field. Finally, records excluded are based on the following reasons duplication, GPU-based implementation, and simply implementing an application using an FPGA.

## BACKGROUND

In the past few years, deep learning has succeeded in artificial intelligence and machine learning, bringing huge progress to society. In many machine learning tasks, such as image classification, video processing, speech recognition, language understanding, data is usually represented in Euclidean space. However, in more and more applications, data are

generated from non-Euclidean space and represented as graphs with complex interdependencies between objects (*Wu et al., 2021*). There has been interest in deep learning techniques that can model graph-structured data (*Battaglia et al., 2018*; *Bronstein et al., 2017*; *Gao et al., 2020*; *Geng et al., 2019*; *Zhang, Cui & Zhu, 2020*). GNNs have grown rapidly due to their ability to learn and model from graph-structured data. Early research was mainly on Recurrent Graph Neural Networks (RecGNNs) (*Sperduti & Starita, 1997*; *Scarselli et al., 2009*; *Gallicchio & Micheli, 2010*), which learn the representation of target nodes by iteratively propagating neighbor information until a stable fixed point is reached (*Zhou et al., 2020*).

With the rapid development of CNN, deep learning has been taken to a new level. CNN's translation invariance, locality, and compositionality make it suitable for processing Euclidean Structured Data such as images, and it can also be applied to various other fields of machine learning. One of the reasons why deep learning is successful is that we are able to extract valid data from Euclidean data. It hinders the transformation of CNN from Euclidean space to non-Euclidean space due to the difficulty of defining local convolutional filters and pooling operators. Extending deep neural models to non-Euclidean space has become an emerging field of research.

Inspired by the success of CNN in the field of deep learning, a large number of neural networks based on convolutional operations have been developed. For example, GCN uses a convolutional neural network to learn Node Features, GraphSAGE uses neighborhood sampling to implement inductive learning on large-scale data sets, and GAT uses an attention mechanism to obtain the weight of neighborhood nodes. They both contain GCNs with convolutional operations, and GCN is the core of building other models. Algorithmic research on GCNs has been extensive (*Wu et al., 2020*; *Abadal et al., 2021*; *Wu et al., 2021*), but there are some challenges in applying it to new applications and demonstrating its efficiency. Due to these factors, the development of the field of GCNs appears to have reached a turning point, and how to achieve the efficient inference of GCNs has become an important research theme to realize its full potential.

Although GCNs have shown good inference results, their inference process is still high cost in terms of latency, computational resources, and energy consumption. An existing popular and effective solution is to design a specialized accelerator for a specific domain, which can solve the inefficiency of the existing architecture because it can customize the hierarchical structure and computing units according to the specific workload (*Wang et al., 2019*). Because of the characteristics of GCNs, they can be optimized from the following aspects. First, the aggregation phase needs to work hard to alleviate memory access irregularities caused by the unbalanced number of neighbors, which mainly relies on graph preprocessing and an efficient and load-balanced sparse matrix processing architecture. Second, the combination phase is like a fully connected layer of a neural network, which requires more use of regularity to improve intensive computation with multiple levels of parallelism. Third, execution order and model quantization are also optimizable parts when designing accelerators. However, many existing structures fail to meet these needs resulting in inefficiencies. On the CPU, the irregularity of the aggregation phase makes GCNs unsuitable for current cache hierarchy designs and data prefetching techniques.

Furthermore, it is difficult for the CPU to efficiently utilize highly reusable parametric data between computational units (*Chen, Emer & Sze, 2016*). And GPU is inherently optimized for compute-intensive workloads with regular execution patterns, such as neural networks. But GPUs are inefficient at processing aggregation phases with irregular memory accesses. Furthermore, combinatorial processing with strong parameter sharing also requires expensive data copying and thread synchronization (*Lindholm et al., 2008*).

In addition to CPU and GPU, FPGA is emerging as a candidate platform for neural network processing (*Guo et al., 2017*; *Mittal, 2020*). FPGA can realize high parallelism and simplify logic according to the calculation process of the neural network, combined with the hardware design of a specific model. Recently, FPGA-based inference models have achieved performance and power consumption improvements of dozens or even thousands of times over CPUs and GPUs. Therefore, FPGAs can achieve higher energy efficiency than CPU and GPU. FPGAs, which combine fine-grained computing, high parallelism and programmability, are ideal for customizing accelerators for GCNs.

There have been some investigations on GNNs. At the algorithm level, *Bronstein et al. (2017)* outline deep learning methods in the non-Euclidean space, which is the first review of GNN, mainly on graph neural networks that include convolutional layers. *Lee et al. (2019)* conducted a partial survey of GNNs applying different attention mechanisms. *Hamilton, Ying & Leskovec (2018)* investigated a limited number of GNNs to analyze how to solve the problem of network embedding. But these works are all done at the level of the neural network model. *Geng et al. (2021a)* summarized four types of irregular behaviors in the processing of neural network models, but their work is not specific to GNNs and the computational process of the GNN algorithms are not presented. Regarding hardware acceleration, *Abadal et al. (2021)* conducted a more comprehensive survey of GNN from a computational perspective, conducted an in-depth analysis of current software and hardware acceleration schemes, revealed the emerging field of GNN accelerators, and elaborated on existing challenges and opportunities. However, there is no in-depth analysis of FPGA based accelerators. As evidenced by, first, the lack of a more in-depth analysis of their unique computing architecture, second, the lack of a summary of the challenges of implementing GNN accelerators on FPGA platforms. Currently, there are still many challenges in using FPGA to accelerate GCNs, including efficient processing of sparse matrices, load imbalance, differences in computing modes, and quantization and maintaining model accuracy. To facilitate the follow-up research work, our work helps the readers to understand the computational process of these accelerators by presenting the computational process of several representative GNNs algorithms first. What's more,we conduct a comprehensive review of existing FPGA-based accelerators for GCNs and review some design details of the accelerators from the perspective of the above four challenges.

## GNN AND GCNS MODELS

GCN learns node features by defining convolution operations in GNN, GraphSAGE uses neighbor sampling to enable GNN to adapt to large-scale datasets, and GAT takes the attention mechanism in transformer to learn edge information. Check Table 2 for their

**Table 2 The main features of different GNN models.**

| GNN models | Main features |
| --- | --- |
| GNN (*Scarselli et al., 2009*) | Fixed-point iteration method |
| | The same parameters are used in feature aggregation |
| GCN (*Bruna et al., 2013*) | Aggregation based on degree matrix and adjacency matrix |
| | More advanced operations to extract node information |
| GraphSAGE (*Hamilton, Ying & Leskovec, 2017*) | Mini batch training |
| | Three aggregator, mean, LSTM and pooling |
| GAT (*Veličković et al., 2017*) | Adding attention mechanism of transformer |
| | More interpretable information |

characteristics. This section will introduce the traditional GNN model and several representative GCNs models above.

## GNN

GNN is a deep learning method that operates on the graph domain; it has been successful in many applications, such as molecule property prediction (*Fout et al., 2017*), recommender systems (*Fan et al., 2019*), traffic speed prediction (*Xie et al., 2020*), computer vision (*Wang et al., 2018*), particle physics (*Ju et al., 2020*), and resource allocation in computer networks (*Rusek & Chołda, 2018*) already utilize GNNs to accomplish their tasks.

Given is a graph G, there are multiple nodes in the graph, and each node and the edge connecting two nodes has its characteristics. The learning goal of GNN is to obtain the hidden state of each node. For each node, its hidden state needs to contain information from neighbor nodes, so the information of neighbor nodes needs to be aggregated to the target node. GNN does this by iteratively updating the hidden state of all nodes.

First, we have a hidden state update function f that is shared among all nodes, also called a local update function, which can be represented by Eq. (1).

$$h_u = f\left(x_u, x_{(e[u])}, h_{(n[u])}, x_{(n[u])}\right) \tag{1}$$

where, $x_u$ refers to the feature of node u itself, and $x_{e[u]}$ represents the features of the edges associated with node u, $x_{n[u]}$ represents the neighbor node features of node u, $h_{n[u]}$ represents the hidden state of the neighbor node of node u at the current moment.

In Fig. 1, a simple graph structure with six nodes was given and represented the edges connecting them. We focus on this local area containing node one and its two neighbor nodes. Then for node 1, its hidden state update function can be expressed by Eq. (2) as:

$$h_1 = f\left(x_1, x_{(1,2)}, x_{(1,3)}, h_2, h_3, x_2, x_3\right) \tag{2}$$

Using the update function, we can continuously use the hidden state of the neighbor node at the current moment,they are part of the input which will be used to generate the hidden state of the target node at the next moment until the hidden state of each node

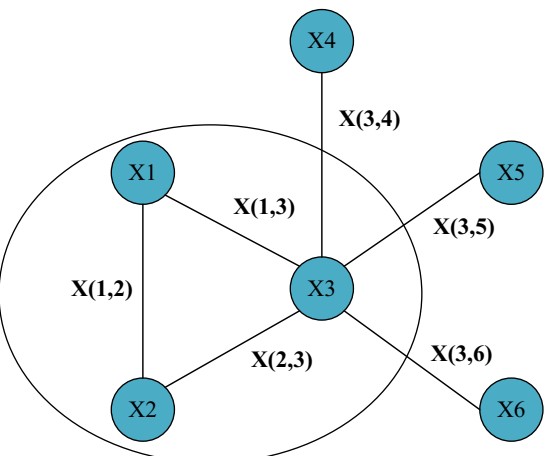

**Figure 1 Simple graph structure with six nodes and the lines between nodes represent information about the edges.**

changes very little. If we use F to denote the function obtained by stacking all the local update functions f, that is, the global update function, then the state update function of all nodes on the graph can be expressed by a more compact Eq. (3).

$$H^{t+1} = F(H^t, X) \tag{3}$$

At this time, as long as F is a compressed map, according to the fixed point theorem, $H_0$ will converge to a fixed point after continuous iteration, which is called a fixed point.

In the classic GNN model, the way to ensure that F is a compression map is to use a feedforward neural network to simply splice the features of each neighbor node, the hidden state, the features of each connected edge, and the features of the node itself. Together, do a simple summation after going through the feedforward neural network.

However, this state update of GNN is not one-step but based on a general framework, Message Passing Neural Network (MPNN) (*Gilmer et al., 2020*). The basic idea is as follows: the vectors representing nodes are obtained after k rounds of message propagation mechanism iteration through the message function M (message) and the update function U (update). For the convenience of description and understanding, we divide the state update process of GNN into the aggregation phase and combination process, and the corresponding functions are aggregation function: Aggregation (Agg) and Combination (Com). As shown in Eqs. (4) and (5), we can express the forward propagation of GNN in the k layer as:

$$a_u^{k+1} = Agg\left(h_v^k \, v \in N_u\right) \tag{4}$$
$$h_u^{k+1} = Com\left(a_u^{k+1}\right) \tag{5}$$

$a_u^{k+1}$ is the aggregated feature of the node u at the k+1th layer, and $h_u^{k+1}$ is the updated output feature of the node u of the kth layer. As shown in Fig. 2, the aggregation function collects the neighborhood features of the target node $U_1$, and the combination function transforms the features of the node $U_1$ through the neural network.

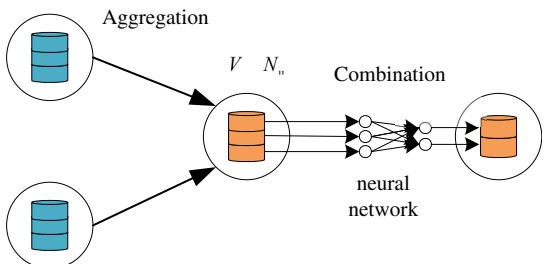

**Figure 2** **The hybrid computing paradigm of GNN which includes combination and aggregation.**

Although the GNN model has shown the potential to handle graph data, but it has some
limitations. On the one hand, using an iterative approach to update node features for fixed
points is inefficient. On the other hand, the original GNN uses the same parameters in
feature extraction, and the model cannot learn deeper feature representations. Therefore,
the variant model of GNN emerges as the times require. GCNs can use different
parameters in different network layers to perform hierarchical feature extraction. Several
representative GCNs models, such as GCN, GraphSAGE, and GAT, are illustrated below.

### GCN

GCN (*Bruna et al., 2013*) extracts node features of graph data by utilizing convolution
operations, which is similar to a feature extractor and has been used in many applications
successfully (*Zhao et al., 2020*; *Han et al., 2019*). Traditional GNN uses the same parameter
to aggregate neighbor information in the aggregation phase. In the GCN model, the
convolution operation allows the aggregation phase to selectively extract neighbor
information rather than a simple summation.

As shown in Eq. (6), we first consider a multi-layer graph convolutional network whose
layer-to-layer propagation rules are as follows:

$$H^{(k+1)} = \sigma\left(\tilde{D}^{-\frac{1}{2}} \tilde{A} \widetilde{D}^{-\frac{1}{2}} H^{(k)} W^{(k)}\right) \tag{6}$$

where, $\tilde{A}$ represents the adjacency matrix, including self-connection in the undirected
graph, and $\tilde{A} = A + I$, $I$ represents the identity matrix because we want to preserve the
feature information of the node itself when the node updates the information. $H^{(k)}$
represents the feature matrix of the kth layer, $W^{(k)}$ is a trainable neural network weight
matrix, $\tilde{D}$ is the degree matrix of the node, where $\tilde{D}_{ii} = \sum_j \tilde{A}_{ij}$ is used to represent the
distribution density of node neighbors. Each layer of GCN is multiplied by the adjacency
matrix $\tilde{A}$ and feature matrix $H^{(k)}$ to obtain a summary of the neighbor features of each
vertex and then multiply by a weight matrix $W^{(k)}$, through the activation function $\sigma$ to do a
nonlinear transformation obtains a matrix $H^{(k+1)}$ that aggregates the features of neighbor
vertices. The normalization operation $\tilde{D}^{-\frac{1}{2}} \tilde{A} \widetilde{D}^{-\frac{1}{2}}$ on the neighbor matrix $\tilde{A}$ is to maintain
the original distribution of the feature matrix in the information transmission process,
preventing some high-degree and low-degree vertices from producing large differences in
feature distribution.

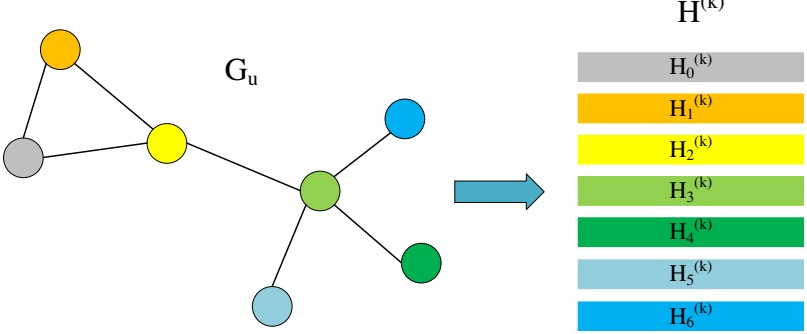

**Figure 3** Node information matrix and each row represents the feature vector of a node.

**Figure 4** The process of node information update, the first stage represents the aggregation process and the second stage represents the combination process.

When we only focus on $\tilde{A}H^{(k)}$, we can find that this is actually a process of aggregating neighbor information. As shown in Fig. 3, we divide $H^{(k)}$ into multiple lines, each line representing is the information of the corresponding node in the graph.

At the same time, considering the adjacency matrix $\tilde{A}$, it is shown in Fig. 4. According to the multiplication rule of the matrix, we can observe that the information update of node 0 needs to aggregate the information of node 0, node 1, and node 2. But this aggregation method is not reasonable enough because it just does a simple addition of neighbor information. If a neighbor node has many adjacent nodes, its correlation with the target node is not strong enough, so the information it transmits to the target node should be multiplied by a corresponding ratio. This is also the meaning of the normalization operation $\tilde{D}^{-\frac{1}{2}}\tilde{A}\tilde{D}^{-\frac{1}{2}}$. $\tilde{D}$ sums up each row of $\tilde{A}$. After normalization, the information of neighbor nodes will participate in the aggregation in a corresponding proportion, and this proportion is related to the degree of neighbors. In this way, the aggregation phase of the neighbor information is completed and then multiplied by a weight matrix $W^{(k)}$, and a nonlinear transformation is performed by the activation function $\sigma$ to obtain the matrix $H^{(k+1)}$, which completes a feature update, also called the combination process. Actually, the process of combination and aggregation can be reversed, which will be discussed in the later section, Execution Order.

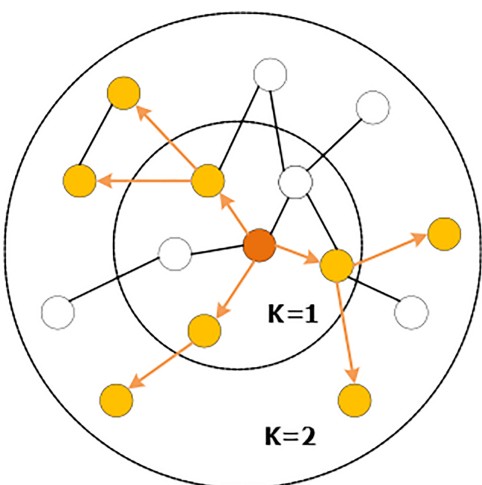

**Figure 5  Random sampling of neighboring nodes.**

After all the nodes complete the information update, a layer of graph convolution network is implemented. Repeat the above process k times to obtain a multi-layer graph convolution network, and obtain the final $H^{(k)}$ as a node representation, it is sent to the corresponding downstream task to realize other functions, such as node classification.

## GraphSAGE

On the one hand, GraphSAGE (*Hamilton, Ying & Leskovec, 2017*) transforms GCN from a full batch training method to a node-centered mini-batch training method by sampling neighbors, avoiding the problem of the neighbor explosion so that it can be used on large scales. Inductive learning is implemented on large-scale datasets, and on the other hand, the algorithm expands the operation of aggregating neighbor information.

Since the degree of some nodes in a large graph will be very large, the time cost of traversing the subgraph, the computational cost of model training, and the storage cost will become uncontrollable. To this end, GraphSAGE uses the operation of sampling neighbors to control the growth rate of nodes as the subgraph diverges.

The sampling operation is defined by setting the sampling depth k and the sampling size s. As shown in Fig. 5, starting from the central node, the first-order (1-hop) neighbors are sampled, and the sampling scale is $S_i = 3$, and then each first-order neighbor is used as the starting point to sample the second-order (2-hop) neighbors. For sampling, the sampling scale is $s_i = 2$, and the space complexity of sampling is fixed at $O\left(\prod_{i=1}^{k} s_i\right)$. This can release a certain amount of storage and reduce the amount of computation when dealing with large graphs. With the increase of the value of k, the computational cost will also increase exponentially, which leads to the fact that the algorithm cannot have a too deep structure, but the experiments show that GraphSAGE can already show high performance when k = 2.

GraphSAGE investigates the properties required for aggregation. On the one hand, aggregation must be adaptive to the number of aggregation nodes. No matter how the

number of neighbors of a node changes, the dimensions of the output after the aggregation operation must be consistent, which is generally a vector of uniform length. On the other hand, the aggregation has arrangement invariance to aggregation nodes, which requires that regardless of the neighbor nodes, the output result is always the same. From the perspective of model optimization, the aggregation must also be derivable. With the guarantee of the above properties, aggregation can be adaptive to any set of input nodes. After comparing three aggregation functions (mean aggregator, LSTM aggregator, and pooling aggregator), it is found that the aggregation functions of LSTM and pooling-based are more profitable than mean and GCN-based. However, LSTM is designed for ordered data rather than unordered data, and the pooling-based aggregation function maintains an advantage in latency.

GraphSAGE deconstructs the GCN from the perspective of airspace, introduces the step of sampling node neighbors, and compares and analyzes the performance of several different aggregation functions. It not only reduces the calculation amount of the model and shows strong performance but also improves the engineering value of the algorithm, so this method has been successfully applied to industrial-scale large-scale recommendation systems, and the effect is very significant (*Lee et al., 2019*).

## GAT

The graph attention network (GAT) (*Geng et al., 2021a*) is based on GCN, adds the attention mechanism in the transformer, and the importance of each neighbor node to the target node in the aggregation phase is represented by calculating the attention coefficient.

Each layer of the GAT model has the same structure, called a graph attention layer. The input of each layer is a set of node features, $H = H_1, H_2, \ldots, H_N$, N represents the number of nodes, and the output of each layer is a new set of node features $H' = H'_1, H'_2, \ldots, H'_N$. The process from H to $H'$ needs to go through multiple steps. First, the input node features need to undergo a learnable linear transformation. Therefore, as shown in Eq. (7), a shared linear transformation is used for each node with a weight matrix W:

$$H' = HW \tag{7}$$

As shown in Eq. (8),Then use a shared attention mechanism a to calculate the attention coefficient $e_{ij}$ of a neighbor node $j$ for the target node $i$:

$$e_{ij} = a\left(WH_i, WH_j\right) \tag{8}$$

This reflects the importance of node $j$ to node $i$, where $j \in N_i$, $N_i$ is the set of first-order neighbor nodes for node $i$, including node $i$ itself. $a$ is a single-layer feedforward neural network incorporating a nonlinear variation of LeakyReLU (negative input slope $\alpha = 0.2$). As shown in Eq. (9),To make the coefficients easy to compare across different nodes, the softmax function is used to normalize all neighbor nodes $j$:

$$\alpha_{ij} = softmax_j(e_{ij}) = \frac{\exp(e_{ij})}{\sum_{k \in N_i} \exp(e_{ik})} \tag{9}$$

This normalized attention coefficient is then used to extract high-level representations of neighbor features in the aggregation phase, applying a nonlinear activation function $\sigma$ to generate output features for each node at that layer, as shown in Eq. (10):

$$H_i' = \sigma \left( \sum_{j \in N_i} \alpha_{ij} W H_j \right) \tag{10}$$

GAT extends the attention mechanism to multi-head attention to refine this learning process, as shown in Fig. 6. GAT uses k independent attention mechanisms to implement the feature update process described above and then concatenates the resulting features, as shown in Eq. (11):

$$H_i' = \bigcup_{k=1}^{K} \sigma \left( \sum_{j \in N_i} \alpha_{ij}^k W^k H_j \right) \tag{11}$$

U represents connection, $\alpha_{ij}^k$ represents the kth independent attention coefficient, $W^k$ represents the weight matrix of the linear transformation corresponding to the kth independent attention mechanism, which is the process of combining feature updates under the kth independent attention mechanism. It is worth mentioning that if this multi-head attention mechanism is used on the output layer of the network, the average method can be used instead of the connection, and then the activation function can be applied for nonlinear transformation, as shown in Eq. (12):

$$H_i' = \sigma \left( \frac{1}{K} \sum_{k=1}^{K} \sum_{j \in N_i} \alpha_{ij}^k W^k H_j \right) \tag{12}$$

GAT is computationally efficient, the computation of attention coefficients and output features can be parallelized across edges and across nodes respectively. The computational complexity is similar to that of GCN. Although the multi-head attention mechanism will expand the storage space and calculation parameters to k times the original, the k calculations are completely independent, so parallelism can also be achieved.

Different from GCN, this attention mechanism introduced by GAT will be more advanced than GCN's feature extraction method based on node degree. The attention coefficients obtained from this analysis have higher interpretability, which will make GAT perform better in inference applications, such as the field of machine translation (*Bahdanau, Cho & Bengio, 2014*). The attention mechanism is applied to all edges of the graph in a shared manner, so it does not rely on prior access to the global graph structure or all its node features, making GAT directly applicable to inductive learning.

This section introduces the traditional GNN model and three classic variant models. On the whole, they all include two stages of aggregating neighbor features and feature updates.

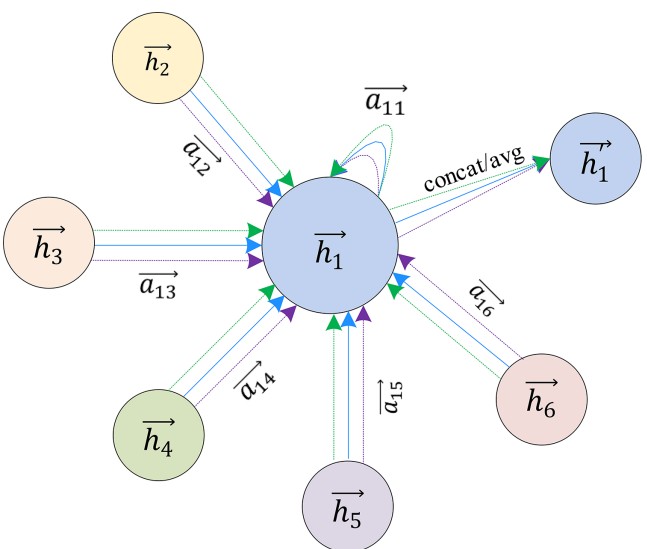

**Figure 6 Multi-head attention used to aggregate neighbor node characteristics.**

The main difference is the way of aggregation when extracting the neighbor feature. A simple summation method is used in the traditional GNN model. GCN uses a node-based degree to represent the proportion of aggregated neighbor information. GraphSAGE proposes three aggregation functions, such as mean, LSTM, and pooling, to extract features from neighbor nodes. The attention mechanism introduced into Transformer by GAT makes the aggregated information more interpretable. In short, deep learning algorithms are continuously proposed to deal with complex graph data, and we do not go into more detail because our work mainly reviews FPGA-based accelerators for GCNs, which will be elaborated on in the next section.

## FPGA BASED HARDWARE ACCELERATORS

There are currently many accelerators under software frameworks such as PyG (*Fey & Lenssen, 2019*), DGL (*Wang et al., 2020*), PCGCN (*Tian et al., 2020*), AliGraph (*Yang, 2019*), AGL (*Zhang et al., 2020*) that simplify the execution of GCNs and achieve significant speedup in model inference. Custom hardware accelerators are a viable way to continue to achieve order-of-magnitude improvements in neural network inference, and this has been achieved on CNN (*Chen, Emer & Sze, 2016*; *Han et al., 2016*; *Kim, Ahn & Yoo, 2017*; *Kim et al., 2017*; *Bai, Zhao & Huang, 2018*). Because of their fine-grained computing, high degree of parallelism, and programmability, FPGAs are a candidate platform for processing neural network inference. However, implementing inference of GCNs on FPGA still needs to overcome some challenges. This section reviews the currently released FPGA-based GCNs accelerators and introduces some details of the above designs from four perspectives: efficient operations on sparse matrix, load balance, execution order, quantify and accuracy.

## Overview

This section reviews the currently released accelerators of GCNs based on FPGA, analyzes the reasons for their success, and collates their characteristics in Table 3.

AWB-GCN (*Geng et al., 2020*) proposed a sparse matrix multiplication (SPMM) kernel that can efficiently handle matrices with power-law distribution, the data in memory is input to a set of processing units (PEs) and accumulators through task distributor and queue (TDQ), and two kinds of TDQ are designed according to data sparsity, TDQ1 is suitable for medium sparsity, TDQ2 is suitable for super sparsity. AWB-GCN achieves dynamic adjustment of workloads between PEs through three hardware-based auto-tuning techniques (distribution smoothing, remote switching, evil row remapping), the details of which will be introduced in "GraphSAGE". These three automatic tuning techniques are the most critical work of AWB-GCN and the main reason for its success.

LW-GCN (*Tao et al., 2021*) proposed a lightweight software-hardware co-optimization accelerator. The software introduced the PCOO matrix compression format to compress input data, which is easy to decompress in hardware. LW-GCN has designed a micro-architecture to handle matrix multiplication, uses optimized computational pipelines in each processing unit to overcome irregularities in memory access while improving data throughput, and is balanced by tiling workload between processing units. In addition, LW-GCN reduces the memory requirements of the model and maintains the accuracy through quantization, and LW-GCN is successful on edge devices with limited resources.

SPA-GCN (*Sohrabizadeh, Chi & Cong, 2022*) is a GCN accelerator specialized for processing small graphs, employing deep pipelines with different levels and degrees of parallelization to improve performance. The author first proposes an infrastructure for processing GCN and then deeply explores the possible parallelism in GCN computations through node-level parallelization, feature-level parallelization, and inter-layer parallelism. And batch processing achieves a breakthrough in performance and maps the optimized architecture into three FPGAs with different configurations. Meanwhile, SPA-GCN accelerates an end-to-end application, SimGNN (*Bai et al., 2019*), with a four-level parallelized efficient architecture, improving the real-time performance of GCN-based graph matching.

FP-GNN (*Tian et al., 2022*) analyzes specifically the impact on non-zero operation, memory usage, and inference time by changing the aggregation and combination order. On this basis, an adaptive GNN accelerator framework (AGA) is proposed. The workflow is optimized, including balancing workloads, feature-level parallelism, and node-level parallelization, enabling flexible execution order and efficient resource utilization. FP-GNN also proposes an adaptive graph partitioning (AGP) strategy, which alleviates the memory bottleneck caused by unaligned memory accesses and redundant source node transfers, and eliminates graph repartitioning overhead between GNN layers.

FPGAN (*Yan, Tong & Zhi, 2020*) is based on FPGA to accelerate the inference process of GAT. FPGAN designs a shift calculation unit for the intensive exp operation in GAT, which eliminates the dependence of computing performance on DSP, and uses an exponential approximation algorithm to fit SoftMax to normalize the attention coefficient.

**Table 3 Overview of FPGA based GCNs accelerators.**

| Name | Main features | Graph size | Algorithms | Baseline |
|---|---|---|---|---|
| AWB-GCN (*Geng et al., 2020*) | Three load balancing techniques. Fine-grained pipelining of aggregation and combination. | Large | GCN | PyG-CPU, PyG-GPU, HyGCN |
| LW-GCN (*Tao et al., 2021*) | Apply data Quantization and workload tiling. Works effectively on resource limited edge devices. | Small | GCN, GraphSAGE | PyG-CPU, PyG-GPU, AWB-GCN |
| SPA-GCN (*Sohrabizadeh, Chi & Cong, 2022*) | Four levels of parallelization. GCN-based graph matching. | Small | GCN, SimGNN | PyG-CPU, PyG-GPU, PyG-CPU |
| FP-GNN (*Tian et al., 2022*) | Support flexible execution order. Adaptive graph partition strategy | Large | GCN, GraphSAGE, GAT | PyG-GPU, HyGCN, BoostGCN |
| FPGAN (*Yan, Tong & Zhi, 2020*) | Accelerate GAT inference. Shift addition unit. SoftMax approximation | Large | GAT | PyG-CPU, PyG-GPU |
| BoostGCN (*Zhang, Kannan & Prasanna, 2021*) | PCFA with 3-D partitioning. Two types of feature update modules. Task scheduling optimization for aggregation and combination | Large | GCN | PyG-CPU, PyG-GPU, DGL-CPU, DGL-GPU, HyGCN |
| I-GCN (*Geng et al., 2021b*) | Graph restructuring algorithm—islandization. Improve data locality. Avoiding redundant aggregation | Large | GCN, GraphSAGE, GIN, GCN | PyG-CPU, PyG-GPU, DGL-CPU, DGL-GPU, HyGCN, AWB-GCN |
| BlockGNN (*Zhou et al., 2021*) | CirCore architecture for matrices computation. Performance and resource model. Reduce the computational complexity of GNNs | Large | GCN, GraphSAGE, GAT, G-GCN, GCN, GIN | HyGCN |
| FlowGNN (*Sarkar et al., 2022*) | Generic GNN acceleration framework. Developed by using high-level synthesis (HLS) | Large | GAT, PNA, DGN, VN | PyG-CPU, PyG-GPU, I-GCN |

FPGAN designed a new data structure to align edges, node features, and weights to align these data to achieve efficient computing. In addition, FPGAN also compresses the model size, quantizes node features, and implements fixed-point calculations.

BoostGCN (*Zhang, Kannan & Prasanna, 2021*) proposed a PCFA scheme for memory constraints, which divides the data into three dimensions: (1) Divide the adjacency matrix into multiple sub-blocks. (2) Cache the source node and the target node, respectively. (3) The input features are divided from the feature dimension, which improves the reusability of on-chip data. BoostGCN has designed a feature aggregation module (FAM) and a feature update module (FUM) to handle the operations of the aggregation and combination phases, respectively. Among them, the feature update module has two architectures, divided into Sparse-FUM and Dense-FUM according to the sparsity of the input feature matrix, which is used to achieve efficient calculation under different matrix densities. Besides, BoostGCN also proposes a task scheduling strategy to balance the workload of the aggregation and combination phases.

The I-GCN (*Geng et al., 2021b*) proposed a new algorithm for graph reconstruction—islandization, which can detect nodes with more neighbors and then use the neighbors of the node as a starting point to divide multiple groups of nodes. The non-zero elements of the adjacency matrix are clustered in this manner. Afterward, aggregates and combinations can be performed in these node groups until all nodes are updated. On the one hand, memory access can be completed in a much smaller region than the original, improving data reuse while avoiding many off-chip memory accesses. On the other hand, nodes in a node group have many common neighbors so pre-aggregation may prevent some redundant operations in the aggregation phase.

BlockGNN (*Zhou et al., 2021*) proposes a pipelined CirCore architecture to compute block circulant matrices efficiently. BlockGNN selected the Reddit dataset to analyze the total computation and algorithm strength of GCN, Graphsage, GAT, and G-GCN in the aggregation and combination phases, respectively. Then a structured compression method using block circulant matrices is proposed to reduce the computational complexity. To efficiently calculate the block circulant matrix, BlockGNN designs a Circore structure with three-stage pipelines and proposes a Performance and Resource Model. It helps determine the number of channels and the parallelism of processing units and other hardware parameters to adapt to the input of the GNN model, ensuring that in the different best performance at the input, which is important for FPGA-based reconfiguration.

FlowGNN (*Sarkar et al., 2022*) proposes a general-purpose GNN acceleration framework using high-level synthesis to deal with the imbalanced development between new GNN algorithms and new accelerators. Unlike previous class-specific GNN model accelerators, FlowGNN supports edge embeddings for widely popular GNN models and can be extended to new models. FlowGNN does not rely on graph preprocessing but builds a message passing architecture common to most GNNs, and designs specific components (such as multi-head self-attention in GAT) for different GNN models to achieve compatibility. At the same time, FlowGNN enables multiple levels of parallelism to drastically improve performance, including node parallelism, edge parallelism, apply parallelism, and scatter parallelism.

**Table 4 Efficient operation of some accelerators which contains data preprocess and efficient architecture.**

| Name | Data preprocess | Efficient architecture |
| --- | --- | --- |
| AWB-GCN (*Geng et al., 2020*) | CSC | Two TDQs for different sparsity |
| LW-GCN (*Tao et al., 2021*) | PCOO | Data replication and row grouping |
| SPA-GCN (*Sohrabizadeh, Chi & Cong, 2022*) | Prune the zeros on-the-fly | Input by column-wise and four levels of parallelization |
| FP-GNN (*Tian et al., 2022*) | CSR | Outer product and mixed execution |
| I-GCN (*Geng et al., 2021b*) | Islandization | Remove redundancy of aggregation |

In this section, we review the currently released accelerators for FPGA-based GCNs and describe their characteristics, which are the main reasons for their success. We will review some of the design details of the above accelerators from the perspective of four challenges.

### Efficient operations on sparse matrix

Large-scale matrix operations accompany the inference process of neural networks. Existing matrix multiplication-oriented accelerators (*Yu et al., 2020b*, *2020a*) usually exploit the structured properties of dense tensors and apply data reuse techniques to improve high performance. However, these techniques do not maintain high efficiency in GCNs because the adjacency matrix in GCNs is usually sparse, random, and irregular due to the difference in node degrees. The aggregation phase in GCNs is embodied in the computation by sparse, dense matrix multiplication (SDMM), which is expressed as the multiplication of the adjacency matrix and the feature matrix and the multiplication of the feature and the weight matrix. The inefficiency of matrix operations will seriously affect the speed of model inference. Compressing data format, overcoming irregular memory access, and configuring computing units to achieve efficient processing of sparse matrices is also key link. This section introduces some design details of AWB-GCN, LW-GCN, SPA-GCN, FP-GNN, I-GCN, and Table 4 presents their key information.

AWB-GCN proposed a new and efficient accelerated geometry algorithm and sparse matrix multiplication kernel (SPMM) for matrices with a power-law distribution, as shown in Fig. 7. SpMMeM buffers the input sparse matrix S from off-chip and provides non-zero elements and relevant indices to TDQ. The DCM buffers the columns of the dense input matrix and broadcasts its elements to TDQ. TDQ assigns tasks to individual PEs. Each PE has two units: a multiply-accumulate unit (MAC) and an address generation unit (AGU) for the generation and forwarding of the resulting address. PEs perform concurrent multiplication of non-zero pairs, accumulation of partial results, and data exchange of ACC buffers. Finally, the ACC buffer caches the partial results of the result matrix C and sends them to the next SpMM engine when the entire column calculation is complete. When storing sparse matrices in CSC format, there are two alternative TDQ designs. When the sparse matrix S is a general sparse matrix (sparsity < 0.75), TDQ-1 replaces the above TDQ; when the sparse matrix S is a super sparse matrix, TDQ-2 replaces TDQ above. Among them, TDQ-1 forwards a certain number of non-zero elements (non-zero elements) to each PE for operation in each cycle. To balance the non-zero element distribution in practice, each PE is equipped with multiple task queues (TQ) to ensure

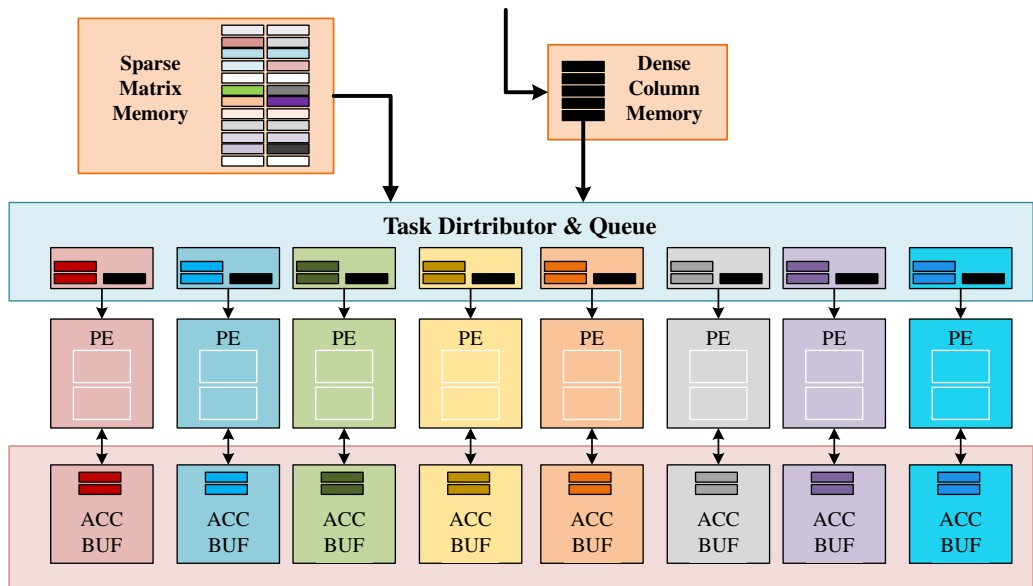

**Figure 7 Architecture of the proposed baseline SpMM engine in AWB-GCN.**

sufficient concurrency to cache all valid data. Before calculation, each element needs to check the Read-after-Write (RaW) risk brought by multiply-accumulate-unit (MAC). RaW risk is detected by checking whether the row index at which the data is calculated is the MAC's current processing row index.

A stall buffer size is set for the delay of the MAC unit to ensure that the danger can be resolved. TDQ-2 uses a multi-stage Omega network to route non-zero elements to the correct PE based on their row indices, solving the problem of highly scattered indices of adjacent elements. TDQ-2 uses a multi-stage Omega network to route non-zero elements to the correct PE based on their row indices, solving the problem of highly scattered indices of adjacent elements. The network is designed to scale better with less hardware complexity. In addition, AWB-GCN makes many effective attempts to balance load work, which will be introduced in "GraphSAGE".

A PCOO format is defined in LW-GCN to compress the inputting sparse matrix, eliminating zero elements to preserve storage space and simplify operations. The PCOO format is also easy to decompress into hardware. LW-GCN also designs a computation engine for efficiently processing multiple non-zero elements.

As shown in Fig. 8, the data of the dense matrix is stored in the dense data memory (DDM). Due to the sparsity and irregularity of sparse matrices, it is difficult to predict the column positions of non-zero elements in advance, which may lead to several PEs that may require different addresses from the same DDM. Limited by the read capability of on-chip memory, this access restriction can lead to data conflicts. To reduce this data conflict, the LW-GCN microarchitecture constructs a multi-port memory through data replication and row grouping. Within the acceptable resource consumption range, r dense data copies are replicated for PE, and each dense data copy is divided into g row groups to reduce the possibility of data conflicts. Due to the additional complexity and resource

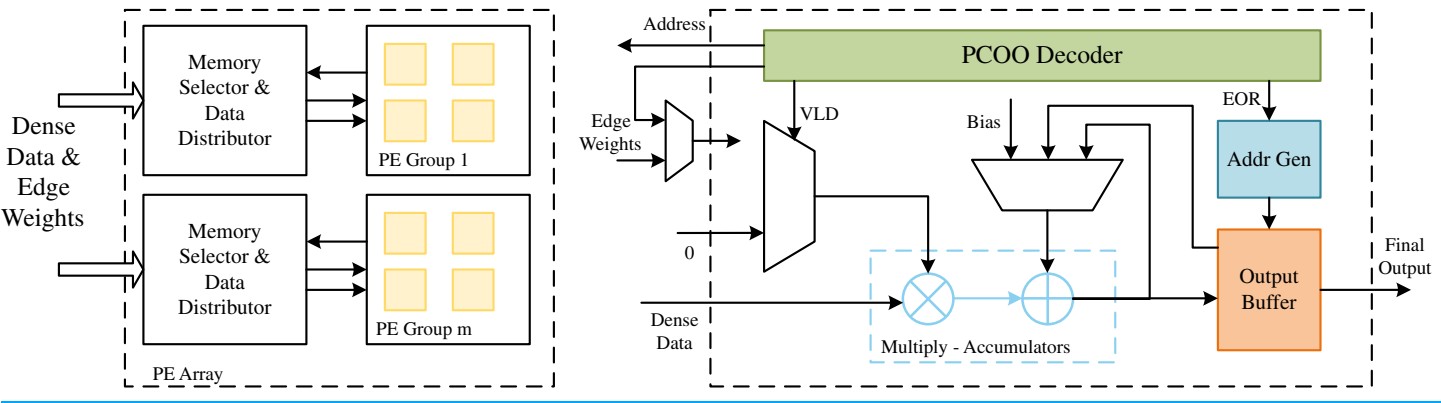

**Figure 8** The architecture of the PE array in LW-GCN.

consumption caused by data replication and a large number of row groups g, LW-GCN conducts experiments with different r and g to determine the optimal number of memory copies and row groups. Based on the address generated by a single PE, the memory selector and data distributor send the corresponding dense data. A priority decoder is used when assigning addresses to memory banks, allowing different PEs to access the same address in the same memory bank. In the SDMM process, the compressed sparse data is directly streamed to each PE. The compressed data is first decoded by the PCOO decoder, and the column index is used as a memory address to obtain dense data. Since multiple rows of data need to be calculated on each PE, SOR and EOR are used to indicate the start and end of a row, respectively. SOR controls the input of the accumulator to its previous result (SOR = 0) or the intermediate result of the previous tile stored in OMMB (SOR = 1). At the same time, the EOR control generates the address used to store the current result in the output buffer and increments the line number of the internal trace (EOR = 1). Finally, the final result is produced by accumulating the results of all tiles. And the underlying SDMM design used by LW-GCN is also applicable to other graph neural network algorithms, such as GraphSAGE, which also achieves very significant results.

SPA-GCN adopts deep pipelines with different levels and degrees of parallelization to improve performance. To avoid RAW dependency, SPA-GCN changes the order of computation, flows into the node information matrix column by column, and reads the weight matrix row by row.SPA-GCN takes an element from an input matrix (read as a stream) and broadcasts it to parallel MAC units. Each MAC unit reads a different element from a pre-stored weight matrix which the information matrix can reuse. This change is depicted in the figure, as shown in Fig. 9, where SPA-GCN divides the workload within the PE by feature-parallelized SIMD operations. To read each element only once, all operations involved in it are completely arranged for each fetched element of $H_l$.

This schedule also increases the cycles before RAW dependency occurs to ensure that different output locations are updated in the next SIMD cycle. The PEs are then replicated by a replication factor (RF), enabling node-level parallelization. The adjacency matrix is usually super sparse when computing matrix multiplications in the aggregation phase.

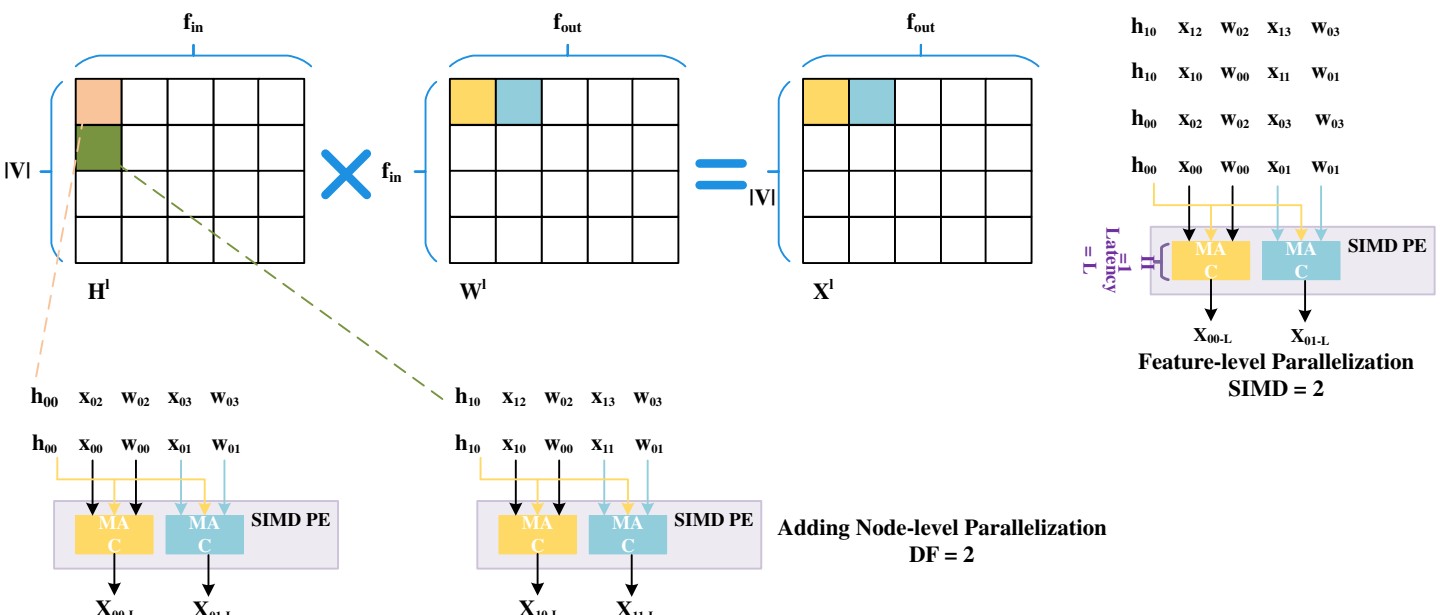

**Figure 9  Feature-level parallelization and node-level parallelization.**

Unlike most accelerators, SPA-GCN does not use on-chip memory to store data structures containing vertices and edges. Rather, the matrix is pruned, and only the non-zero elements representing edges are passed to the FPGA in a stream, and all properties of the target nodes are updated before exiting the edge. To prevent RAW dependency, edges are rearranged during preprocessing of the adjacency matrix so that edges with the same target node are at least L positions apart. This can ensure that no more than one update to the same node is made within L period windows. In this step, SPA-GCN only uses feature-level parallelism to distribute the workload. In addition, SPA-GCN also utilizes a dataflow architecture to connect modules, adding an in-layer pipeline, making the overall latency close to that of the slowest module. At the same time, this operation avoids off-chip memory accesses between modules.

FP-GNN supports different GNN models such as GCN, GraphSAGE, and GAT. FP-GNN designs special PE for them to handle matrix operations. Among them, PE used to calculate GCN and GraphSAGE has a similar structure, while GAT introduced an attention mechanism with more basic operations. First, FP-GNN compresses the sparse matrix in CSR format and merges the index and data to facilitate indexing and save memory space. During the aggregation phase, the adjacency matrix flows into the edge cache (EB) in each processing unit in a compact CSR format, and then the task scheduler assigns edges to each PE array and obtains the corresponding node information from the source node cache (SNC) and the target node cache (NC). The combination phase adopts the outer product method to obtain higher input feature reusability. Since the partial sums are locally accumulated inside each PE, the outer product method avoids data transfer between PEs. Node Features are assigned to the rows of each PE array through a shared

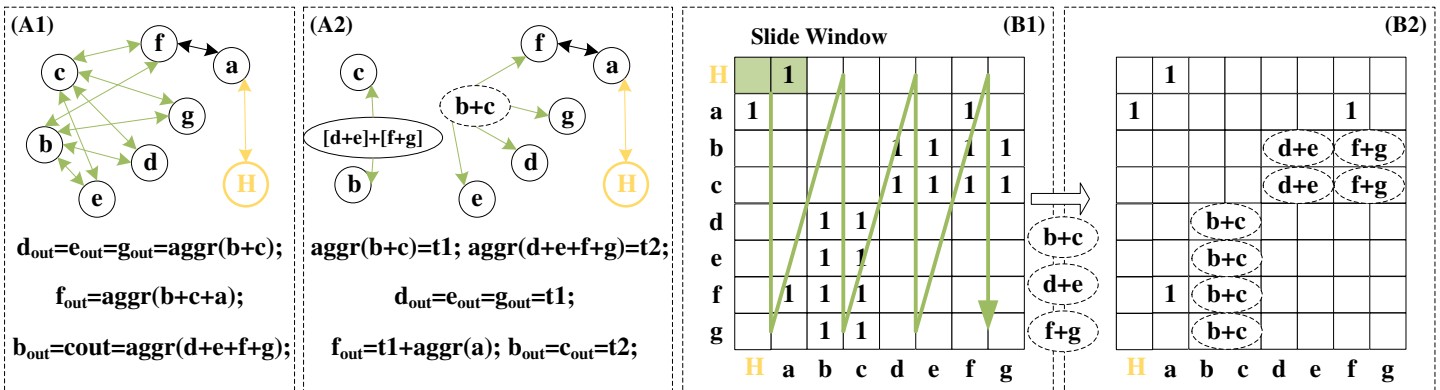

**Figure 10 Redundancy removal of a typical island in I-GCN which can reduce redundant operations in the aggregation process.**

bus, and the weights are also streamed to the PE array columns for corresponding operations. Therefore, the aggregation phase achieves feature-level parallelization by aggregating multiple feature dimensions over the columns of the PE array and exploits node-level parallelism by aggregating multiple target nodes on rows of PE arrays. The combination phase achieves feature-level parallelism by accumulating multiple output feature dimensions on the columns of the PE array and exploits node-level parallelism by converting multiple node features on the rows of the PE array (node-level parallelization).

The graph reconstruction algorithm—islandzation proposed by I-GCN makes the non-zero elements of the sparse adjacency matrix become clustered. This enables higher data reusability when aggregating the information of common neighbors between nodes. Redundant operations in the aggregation phase are avoided. Details of the redundant removal operation are detailed in Fig. 10. A1 and A2 demonstrate redundant operations on common neighbors during the aggregation phase. Nodes d, e, f, and g are the four common neighbors of nodes b and c. When the aggregation is centered on b and c, the eigenvectors of d, e, f, and g are aggregated twice. When the aggregation is centered on d, e, f, and g, the eigenvectors of b and c are aggregated four times. If the feature vector dimension of the node is large, this redundant aggregation will bring great computational complexity. Therefore, two additional virtual nodes are added, and the aggregation results of the precomputed neighbor nodes are given to them, and then they are connected to the actual nodes according to the needs of the aggregation. This precomputed aggregation result can be reused during the aggregation phase.

The data of the public node is aggregated only once, but it participates in the aggregation of multiple nodes. B1 and B2 show examples of searching for common nodes and removing redundant operations in dimension k = 2. Scanning starts when all nodes complete the combination and pre-aggregation of adjacent k nodes. If both positions are 1, it means that the node currently scanned is the common neighbor of the other two nodes. As shown in B1 and B2, d is the common neighbor of b and c, d is no longer repeatedly aggregated but directly uses the pre-aggregation result, reducing a vector addition

**Table 5 Methods for load balance of some accelerators.**

| Name | Methods of load balance |
|---|---|
| AWB-GCN (*Geng et al., 2020*) | Distribution smoothing, remote switching, evil row remapping |
| LW-GCN (*Tao et al., 2021*) | Round-robin assignment |
| SPA-GCN (*Sohrabizadeh, Chi & Cong, 2022*) | Feature-level parallelization |
| BoostGCN (*Zhang, Kannan & Prasanna, 2021*) | Centralized load balancing scheme and phase-level balance |

operation. After the entire adjacency matrix is scanned, the combination and aggregation phase is completed.

## Load balance

Since graph data has different sparsity, the memory location of a node's neighbors and the number of neighbors of each node are irregular, and the degree of a single node generally follows a power-law distribution. This will result in an unbalanced computing workload for each node of the graph, reducing computational efficiency. This is usually manifested in computations where there are large differences in the density of individual rows of the adjacency matrix. Simply dividing the matrix into rows, and assigning each row to a different unit, will result in very different workloads assigned to each unit. The latency of this group of ops will then be dominated by only the densest input rows, which greatly reduces efficiency, and we discuss this challenge separately. In this section, we introduce the work of AWB-GCN, LW-GCN, SPA-GCN, and BoostGCN on workload balancing, and their key information is given in Table 5.

AWB-GCN has made some effective attempts to balance the workload of sparse matrix multiplication computing cores, mainly dealing with load balancing from three aspects, Distribution Smoothing, Remote Switching, and Evil Row Remapping remapping. The Distribution Smoothing structure tracks the number of tasks to be completed in the task queue (TQ) to obtain PE utilization information at runtime and then dispatches the work of those PEs with many pending tasks to those that are relatively less busy. For neighboring PEs, these sent jobs need to be returned and accumulated with the partial results of the original PE after processing. To balance the design complexity, the range of adjacent PEs is set within 3-hop neighbors. However, when non-zero rows are aggregated, the PEs in an area are all busy, and tasks to be completed on PE cannot be sent to adjacent PEs. At this time, the smooth distribution structure will not perform well in balancing the load. Remote switching was proposed to solve the dense row clustering problem. PE Status Monitor (PESM) is used to identify a certain number of overloaded and underloaded PEs. When the number of pending tasks in the TQ reaches 0, a signal is sent to the PESM, which will indicate which PEs are free, save this information in the buffer, then search for the corresponding number of PEs in the overloaded state, and a part of the work of these PEs is exchanged to idle PEs. Since the adjacency matrix is shared in each round of calculation, the current switching strategy is of great significance for the next round of calculation. The

accelerator remembers the switching strategy used in the current round. It is gradually optimized according to the PE utilization information obtained in the next round. Adding remote switching distribution smoothing structures can effectively solve the dense row clustering problem. The main reason for load imbalance is the existence of dense rows. When remote switching cannot handle the huge gap in PEs utilization, Evil Row Remapping will be used to remap the rows that cause PE overload. The task of the overloaded PE will use the Super-PE to switch to a set of Labor-PEs controlled by it in the next round, and the original workload of the Labor-PEs can still exchange tasks with the idlest PEs through remote switching. Super-PEs and Labor-PEs will act as regular PEs if no row remapping is triggered. Experiments show that AWB-GCN achieves 2.11×, 1.41×, 1.62×, 8.75×, and 1.20× PE utilization improvements based on five datasets, respectively.

LW-GCN assigns the multiplication of non-zero elements in a row of an adjacency matrix to the same arithmetic unit (PE), while the multiplication of non-zero elements in different rows is a cyclic way to assign to different PEs. This way, the non-zero elements in each row will be processed sequentially on the same PE, and the same accumulators need not be used at the same time. However, due to large differences in the degrees of nodes in the graph data, different rows of the adjacency matrix may have extremely different densities. If you simply assign each row to a different operation unit, then there is likely to be an inefficient situation. That is most PEs complete operations while waiting for a PE to complete a particularly intensive row operation. As shown in the allocation step in Fig. 11, to improve the efficiency of PEs, each PE is designed to work independently, and each PE starts computing a new row as soon as it finishes the previous PE. Considering that the density of a row is unlikely to be related to the number of rows, according to the Law of Large Numbers, the sum of the densities of the rows assigned to each PE is similar. The experimental results show that the idle time of the PE with the lowest utilization is less than 20% of the SDMM time. However, this round-robin allocation cannot avoid the coincidence that a denser row happens to be allocated to the same PE. Therefore, LW-GCN exhibits a large PE load imbalance in the dataset PubMed, which indicates that there is room for improvement in workload scheduling for this round-robin allocation.

The feature-level parallelization in SPA-GCN can handle workload imbalance well. Unlike other accelerators, SPA-GCN changes the order of computation and reads the node information matrix column by column. Each element is read-only once, and all operations involved are scheduled for each element to read and divide the workload within the PE by feature-parallelized SIMD operations.SPA-GCN adopts a technique of dynamically pruning zero elements, which can skip all operations involving zero node embeddings. Figure 12 shows the benefit of this operation, the non-zero elements of the matrix are remapped to SIMD dimensions by clipping and arbiter, and all CUs in the PE will perform the corresponding valid operation. To ensure correctness, SPA-GCN adds a control unit that tracks the last cycle of each output position update. If the number of cycles between two updates to the same location is less than L, the control unit will insert bubbles in the pipeline until the last update is committed.

BoostGCN utilizes a centralized load balancing scheme to distribute tasks to the FAM. When FAM completes feature aggregation for one partition, it will get another partition

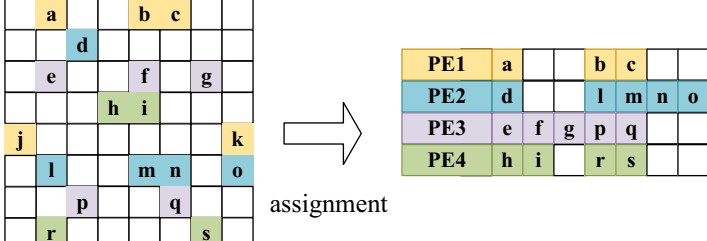

**Figure 11 Round-robin assignment in LW-GCN which tiling workloads to multiple PEs.**

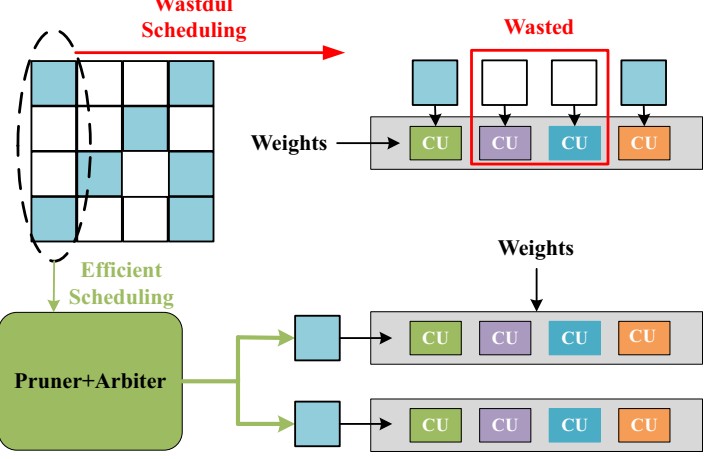

**Figure 12 Efficient scheduling in SPA-GCN which can send non-zero elements to each CU *via* pruner and arbiter.**

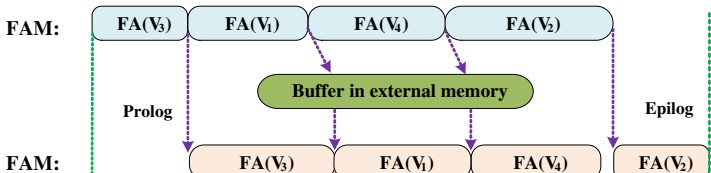

**Figure 13 Task scheduling optimization in BoostGCN which solved the load imbalance between the combination phase and the aggregation phase.**

from the task pool for computation. This solves the load imbalance caused by the uneven distribution of node degrees. Besides, BoostGCN proposes task scheduling to solve the load imbalance problem between the aggregation and combination phases. As shown in Fig. 13, the data is sent to the FAM to perform the calculation in the aggregation phase, and then the aggregated feature vector is sent to the FUM to complete the feature update. Due to the task-level pipelining strategy, if the aggregation phase is processing a partition with many nodes so that the execution time of the aggregation phase is longer than that of the combination phase, this will result in FUM not doing work but waiting for the aggregated feature vector. In order to solve this problem, BoostGCN sorts the partitions according to the number of nodes. FAM first executes the partition with a smaller number of nodes and

uses a buffer to store the aggregated feature vector. FUM can load the aggregated feature vector from this buffer to complete the feature update, which solves the problem of FUM calculation stagnation.

## Execution order

The GNN model is divided into two stages, aggregation and combination, and the executive order does not affect the final result. However, some works have already mentioned the impact of the execution order of these two phases. AWB-GCN analyzes the number of non-zero operations brought by different execution orders of GCN on different datasets. BoostGCN, Engn (*Liang et al., 2020*), GCNAX (*Li et al., 2021*) mentioned that the computation order does not affect the final result, but choosing an appropriate computation order can reduce the number of floating-point operations and external memory accesses. These works all choose to combine first and then aggregation to design accelerators, but this is only observed from the perspective of GCN with fixed feature dimensions. FP-GNN analyzes the effect of changing the execution order under multiple GNN models and multiple feature dimensions, and the accelerator designed on this basis shows excellent performance.

FP-GNN quantitatively analyzes the effect of execution order on non-zero operations, memory footprint, and execution time by changing the aggregation and combination order. Set the execution order with the most operations in each layer as the baseline, *CoAg* and *AgCo* represent the execution order as combination-aggregation and aggregation-combination, respectively. For dense input features, *CoAg* reduces aggregation and its memory footprint more than *AgCo*.For sparse input features, *CoAg* reduces aggregation operations but increases aggregation and aggregation memory footprint. The proportion of aggregation and combination operations are related to the dataset, GNN model, and the number of model layers. Based on the analysis of the above problems, FP-GNN proposes an AGA architecture that supports flexible execution order to handle the aggregation and combination phases and utilizes feature-level parallelization and node-level parallelism and optimization methods such as workload balancing, feature sparsity elimination, and hybrid execution, resulting in good performance and efficiency. Compared with other accelerators, FP-GNN has significant advantages in inference execution time for the four datasets, mainly because the architecture of FP-GNN supports flexible execution order to achieve higher computational efficiency, which is the benefit of quantitatively analyzing the impact of execution order on non-zero operations, memory footprint, and execution time.

## Quantify and accuracy

Quantization is an effective method to improve the computational efficiency of neural networks. Compared with full-precision computation, the fixed-point computation can significantly improve inference speed. It reduces computational and memory overhead by converting model parameters into a low-precision data format with less memory overhead. Continuing to maintain model accuracy after employing multiple model-specific

optimizations is another challenge for GCNs accelerators. LW-GCN and FPGAN describe their quantization strategies.

LW-GCN quantizes the values of all matrices in GCNs to further reduce memory requirements. LW-GCN uses a 4-bit signed integer (SINT4) to perform the quantization of the input feature value. During the computation, store the intermediate result as a 32-bit signed integer SINT32, and after the final result of each layer is obtained, it is performed quantization of a 16-bit signed integer SINT16. The quantization strategy is evaluated on GCN and GraphSAGE of three datasets, and the results show that the accuracy loss caused by using the quantization strategy is controlled within 0.2%, which is almost negligible.

To save memory and reduce the computational difficulty, FPGAN (*Yan, Tong & Zhi, 2020*) compresses the model, and its core idea is to convert the weights to powers of 0 or 2 and judge whether retraining is required by observing the loss of accuracy after conversion. If the compression accuracy loss for one set is within a reasonable range, start the next set of compressions. Otherwise, retraining is required to reduce the accuracy loss. Model compression allows larger models to fit in the original memory space. In FPGAN, the designed shift operation unit is used to reduce the dependence on DSP, and the input feature needs to be mapped from a floating-point number to an integer range before the shift operation. FPGAN first calculates the quantization coefficient $Q^{(l)}$ of each layer through Eq. (13).

$$Q^{(l)} = round\left(\log_2 \frac{2^{\beta-1} - 1}{\max(abs(a^{(l)}))}\right)$$ (13)

Among them, the round is a rounding function, $\beta$ is the number of bits of the quantization feature, and $\max(abs(a^{(l)}))$ is the maximum value of the absolute value of the input feature of this layer. As shown in Eq. (14), the value after quantization can be expressed as:

$$a_{int} = round(2^Q \bullet a_{float})$$ (14)

$a_{float}$ and $a_{int}$ represent the floating-point number before quantization and the integer after quantization, respectively. Experimental results show that the inference results of FPGAN maintain good accuracy compared to the full-precision model pyGAT.

## Performance and discussion

Due to the limited resources on the FPGA, resource consumption is an important reference for evaluating accelerator performance. We give the resource consumption of some accelerators, as shown in Table 6. It should be noted that Intel and Xilinx use ALM and LUT, respectively, for the logic resources of FPGA, which is represented by Logic Resource, uniformly. I-GCN did not give more detailed data, so we did not put it in for comparison. FP-GNN is the most resource-intensive among all accelerators in summary. The reason is that FP-GNN uses resources to support Adaptive Accelerator Architecture (AGA) and Adaptive Graph Partitioning Strategy (AGP), making FP-GNN have stronger algorithm and data adaptability. LW-GCN uses the least logic resource and DSP to achieve

**Table 6 Resource consumption and frequency of accelerators.**

| Accelerator | Device | Logic resource | BRAM | DSP | Frequency (MHz) |
|---|---|---|---|---|---|
| AWB-GCN (*Geng et al., 2020*) | Stratix 10 SX | 700,000/2,800,000 | N/A | 8,192/11,520 | 330 |
| LW-GCN (*Tao et al., 2021*) | Xilinx Kintex-7 K325T | 161,529/326,080 | 291.5/445 | 512/840 | 200 |
| FP-GNN (*Tian et al., 2022*) | Xilinx VCU128 | 717,578/2,852,000 | 1,792/2,016 | 8,192/9,024 | 225 |
| BoostGCN (*Zhang, Kannan & Prasanna, 2021*) | Stratix 10 GX 10M | 389,000/3,466,080 | N/A | 3,584/5,760 | 250 |
| BlockGNN (*Zhou et al., 2021*) | Xilinx ZC706 | 85,254/218,600 | 452/1,090 | 882/900 | 100 |
| FlowGNN (*Sarkar et al., 2022*) | Xilinx Alveo U50 | 229,521/872,000 | 185/1,344 | 1,048/5,925 | 300 |

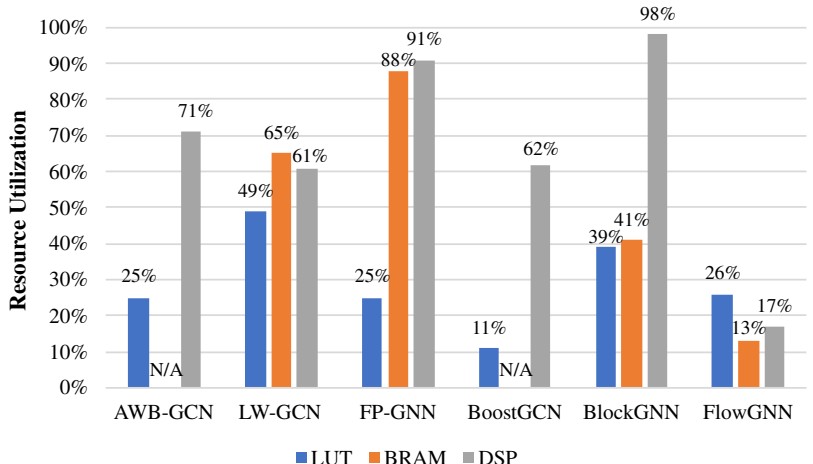

**Figure 14 Hardware resource utilization of accelerators.**

a decent performance improvement, which is of great significance to GCN deployment in edge devices. The best performance in terms of acceleration, FlowGNN, still performs well in terms of resource consumption. This is because the multiple levels of parallelism of FlowGNN make resource utilization more efficient. Figure 14 shows the resource consumption ratio of some accelerators. The DSP is at the heart of the computation, so it deserves a separate discussion. From the perspective of consumption ratio, the DSP utilization rate of BlockGNN is as high as 98%, which greatly affects their frequency. The frequency of BlockGNN has been as low as 100 MHz, which reduces the computing efficiency. However, AWB-GCN relies on a complete three-level task scheduling scheme and reaches the highest frequency of 330 MHz under the condition of high DSP usage.

The above FPGA-based GCNs accelerators have achieved great performance improvements compared to the acceleration under the software framework. As an earlier hardware accelerator, HyGCN (*Yan et al., 2020*) proposed a hardware design with two efficient processing engines, which effectively overcomes the irregularity of the aggregation phase and makes full use of the regularity of the combination phase, and implemented on a 12 nm ASIC. Compared to state-of-the-art software frameworks running on Intel Xeon CPU and NVIDIA V100 GPU, HyGCN achieves an average speedup of 1,509× and 6.5×,

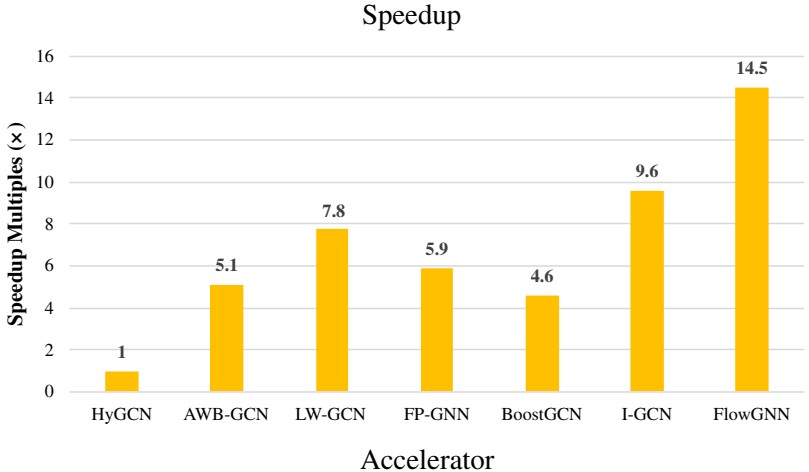

**Figure 15 Speedup of accelerators compared with HyGCN.**

respectively. This also makes HyGCN a target for performance comparisons of other FPGA-based accelerators for GCNs. As an early work on FPGA-based accelerators for GCNs, AWB-GCN achieves phenomenal improvements in performance by relying on three approaches to balance workloads. Compared with HyGCN, AWB-GCN improves inference speed by 5.1 times. We will use HyGCN as the baseline because it was an early work. The latency with different datasets and throughput of accelerators are given in Tables 7 and 8. For those accelerators that are directly compared to HyGCN, we calculate the average speedup for different datasets based on the multiplicative relationship of latency. For others, we represent speedup by an indirect method. For example, FlowGNN was evaluated on two different datasets (Cora, CiteSeer), and normalized the number of DSPs, the evaluation results obtained an average speedup of 2.5 times compared to AWB-GCN. Compared to HyGCN, AWB-GCN achieves an average speedup of 5.8× (based on Cora and CiteSeer). So in Fig. 15, we use 14.5 (2.5 × 5.8) to denote the speedup multiplier of FlowGNN compared to HyGCN. The results are shown in Fig. 15. In summary, all comparisons were made based on their common datasets. Furthermore, we evaluate the throughput of these accelerators uniformly by calculating the number of bits per second of input. It can be easily observed that I-GCN and FlowGNN demonstrate a great advantage in average throughput. It must be mentioned that since SPA-GCN improves the performance of SimGNN, an end-to-end application, FPGAN is only for the GAT algorithm, BlockGNN does not give explicit data of latency, so they were not used for comparison. The data in the figure reflects the average performance improvement of the FPGA-based GCNs accelerator in accelerating GCN compared to HyGCN on common datasets. We can observe that these accelerators all achieve high-performance improvements compared to HyGCN, among which FlowGNN achieves up to 14.5× speedup by relying on multiple levels of parallelism.

**Table 7 Latency of accelerators with different datasets.**

| Accelerator | Cora | CiteSeer | Pubmed | Nell | Reddit |
| --- | --- | --- | --- | --- | --- |
| HyGCN | 21 | 300 | 640 | N/A | 289,000 |
| AWB-GCN | 17 | 29 | 230 | 3,250 | 49,700 |
| LW-GCN | 11 | 17 | 167 | NA | N/A |
| FP-GNN | 36 | 61 | 539 | N/A | 17,100 |
| BoostGCN | 76 | 125 | 1,140 | N/A | 18,850 |
| I-GCN | 8.2 | 12.9 | 110 | 1,200 | 46,000 |
| FlowGNN | 7.8 | 10.4 | N/A | N/A | N/A |

**Table 8 Average throughput with different datasets of accelerators.**

| Accelerator | HyGCN | AWB-GCN | LW-GCN | FP-GNN | BoostGCN | I-GCN | FlowGNN |
| --- | --- | --- | --- | --- | --- | --- | --- |
| Throughput (Gb/s) | 1,887.2 | 12,104.1 | 1,475.6 | 2,619.8 | 1,291.1 | 30,465.1 | 26,199.6 |

## CONCLUSION AND DISCUSSION

### Conclusion

GCNs have been widely used for graph data processing in recent years, but their target applications often impose severe constraints on latency and throughput. To address this challenge, research on FPGA-based accelerators for GCNs has increased, and many accelerators have overcome many irregularities in processing graph data and achieved orders of magnitude performance improvements. In this article, we reviewed representative GCNs algorithms and FPGA-based GCNs accelerators, summarized their characteristics and compared their performance, and introduced some design details according to different challenges.

In a word, the efficient processing of sparse matrix is the key to speeding up the inference process of GCNs. Balancing the workload can greatly improve the utilization of the computing unit, and selecting the appropriate execution order according to different inputs can reduce the computational cost. Complexity and model quantization can alleviate memory requirements while maintaining accuracy, which is why the above accelerators achieve orders of magnitude performance improvement.

### Discussion

It is foreseeable that in the future, the graph data will be larger, which will continue to challenge the design of accelerators. At present, high-performance accelerators adopt software and hardware co-design, use corresponding software algorithms to partition or compress data formats, and customize an efficient computing architecture to achieve fine-grained computing and more efficient task scheduling. We believe that the future accelerator design will also adopt a co-design scheme to accelerate GCNs inference. However, what needs to be improved is to reduce the complex operations and memory requirements brought about by data preprocessing and even not require data

preprocessing. In addition, the current GCNs computations are all around matrix computations without exception, and a new understanding of graph data may lead to innovative computational forms.

The development speed of GCNs algorithms is faster than that of GCNs accelerators, and this unbalanced development will make maintaining generality a potential feature of future GCNs accelerators. For example, the latest FP-GNN, Flow-GNN, *etc.*, have been able to support more GCNs algorithms than the previous accelerators. Based on this, we propose an outlook for two potential development directions. First, a unified and efficient architecture may emerge in the future to support continuously updated GCNs algorithms. This challenges the adaptability of accelerators. The operation unit is modularized, which is divided into general modules and special modules. Through the transformation of the data path between general modules and the scheduling of special modules to support different GCNs algorithms. Data manipulation strategies, that is according to the input graph data to adjust the corresponding calculation strategy, such as adjusting the execution order and quantization scheme, improve efficiency while maintaining accuracy, and get rid of the dependence on data. Second, the development method based on HDL is an important reason for the imbalance between the speed of the algorithm and accelerator development. Development tools using high-level languages (such as HLS) may become a balanced bridge across this gap. Excellent high-level synthesis tools can ensure that the advantages of software development can be integrated, the learning cost of hardware developers can be reduced, and the work efficiency of accelerator development can be fully released under the premise of meeting design requirements. For example, when designing with HLS, each component can be simulated at the RTL level using the C models of the other components, and can easily take advantage of both coarse-grained and fine-grained parallelism. This allows designers to focus more on the high-level algorithm and architecture design without worrying about low-level implementation details (*Cong et al., 2022*).

In summary, FPGA-based GCNs accelerators will develop in the following directions: software and hardware co-design, efficient task scheduling, higher generality, and faster development speed.

### Funding
The authors received no funding for this work.

### Competing Interests
Ruiqi Chen is a visiting researcher for VeriMake Innovation Lab of Nanjing Renmian Integrated Circuit Co., Ltd.

## Author Contributions

- Shun Li conceived and designed the experiments, performed the experiments, analyzed the data, performed the computation work, prepared figures and/or tables, authored or reviewed drafts of the article, and approved the final draft.
- Yuxuan Tao conceived and designed the experiments, performed the experiments, prepared figures and/or tables, and approved the final draft.
- Enhao Tang conceived and designed the experiments, prepared figures and/or tables, and approved the final draft.
- Ting Xie performed the computation work, authored or reviewed drafts of the article, and approved the final draft.
- Ruiqi Chen conceived and designed the experiments, analyzed the data, prepared figures and/or tables, authored or reviewed drafts of the article, and approved the final draft.

## Data Availability

There is no data or code; this is a literature review.

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
