# Peer review of "A survey of field programmable gate array (FPGA)-based graph convolutional neural network accelerators: challenges and opportunities"

_PeerJ Computer Science, doi:10.7717/peerj-cs.1166_

## Round 0.1 · original submission · Minor Revisions

Please consider the reviewers comments when making your revisions.

Reviewer 1 ·

Basic reporting

This paper lists the recent advances in developing an FPGA-based accelerator for the GCN algorithm. I found the paper to be informative and I believe there is value in publishing it. Nevertheless, I'd like to suggest some improvements with the sole purpose of enhancing the quality of the paper. First, it needs to go under extensive revision with respect to its writing to increase its readability. The paper is filled with grammatical and language errors including, missing subject/verb in a sentence, the wrong place to use a capital letter, ambiguous sentences, repeated sentences, long sentences which make it hard to follow, poor choice of words, etc. (e.g., lines 104, 109, 110, 197 to 199, 218, 238, 381, 523, 535, 782).

I liked the fact that the paper mentions the different acceleration opportunities and discusses how each previous work exploited them. This, in my opinion, is a great addition to the other survey papers. However, Lines 185 to 188 seem to be suggesting that the content of this paper is a subset of the previous 2 survey papers. It will help the paper a lot if you emphasize more on the things that were overlooked in the previous survey papers and the information the reader can expect to get when going through your paper.

With regards to making the paper more readable, it would be nice to make the figures' captions and tables' titles more inclusive so that the readers can understand what purpose they serve by just looking at them. Also, some notations/terminologies need improvement. For example, Eq. 1 uses e[u] for showing an edge of node "u", however, an edge needs to be specified by two nodes. Eq. 3 is missing the graph relations. Line 251 defines the feature extractor as the convolution operation of GCN however, lines 356 and 246 refer to it as the aggregation step. Although GCN is inspired by CNN and uses convolutional operations, it is not accurate to say GCN uses CNN (Line 38). It's better to use consistent notations, for example, either use kth or Kth, but not both. Line 530 uses DF for the acronym of replication factor, it should either be RF or duplication factor.

Experimental design

It's worthwhile to double check all the citations to make sure they are placed correctly. For example, Chen et al (2016) in Line 233 refers to the Eyeriss paper which is a CNN accelerator. Its usage there is not clear.

Validity of the findings

The different accelerators that are compared in Figure 15 do not measure the performance like each other. For example, AWB-GCN uses a hardware counter which measures only the runtime of the kernel, but FlowGNN uses the end-to-end on-board results which include the DMA loads and API overheads as well. Are these differences considered in getting the speedup numbers? It would be nice to explain how these numbers are produced.

·

Basic reporting

The introduction makes it clear that the focus of this survey is on FPGA-based accelerators for Graph Convolutional Neural Networks (GCN). The motivation is supported by examples of applications for GCNs and by listing the challenges of hardware acceleration for this type of networks. The field has been reviewed recently, however this paper provides a more in-depth overview of the GCN FPGA-based accelerators. Overall, clear professional English is used, however there are some minor English flaws that could be easily fixed. For example (but not only), lines 108, 173-175, 218, 244 and 782 are ambiguous. Also, the way figures and tables are referenced should be rechecked. For example, in line 256 Table 6 is mentioned though it is not relevant to the context.

Experimental design

The survey is comprehensive and well organized. The overview of the most representative GCNs and their characteristics is useful and the section is well placed before the analysis and comparison of the FPGA-based accelerators. There are multiple mentions of datasets as a general term and a few are named throughout the text, but it would be useful to know what the exact datasets are the GCNs trained and evaluated on and if the accelerators comparison is done for the same GCN trained on the same dataset.

Validity of the findings

The “FPGA based Hardware Accelerators” section comments on the performance of the accelerators. However, it is done in a vague manner. I suggest adding the performance indicators, such as throughput, latency, resource consumption, when discussing performance. Table 6 summarizes well the resource consumption and frequency of the accelerators. If possible, I suggest adding here the throughput and the latency as well.
The conclusions are well stated and they identify future development directions.

Additional comments

No comment.

---

## Round 0.2 · accepted · Accept

Thank you for addressing the issues in a previous version of your manuscript.

Reviewer 1 ·

Basic reporting

no comment

Experimental design

no comment

Validity of the findings

no comment

Additional comments

The reviewed manuscript addressed my previous concerns. I do not have further suggestions.

·

Basic reporting

The authors have addressed the comments from the first review and made the respective changes.

Experimental design

-

Validity of the findings

-

Additional comments

-